# Flexible and stretchable metal oxide nanofiber networks for multimodal and monolithically integrated wearable electronics

Binghao Wang [1,8], Anish Thukral[2,8], Zhaoqian Xie [3,4], Limei Liu [5], Xinan Zhang[1,6], Wei Huang[1], Xinge Yu [3✉], Cunjiang Yu [2✉], Tobin J. Marks [1,5✉] & Antonio Facchetti [1,7✉]

Fiber-based electronics enabling lightweight and mechanically flexible/stretchable functions are desirable for numerous e-textile/e-skin optoelectronic applications. These wearable devices require low-cost manufacturing, high reliability, multifunctionality and long-term stability. Here, we report the preparation of representative classes of 3D-inorganic nanofiber network (FN) films by a blow-spinning technique, including semiconducting indium-gallium-zinc oxide (IGZO) and copper oxide, as well as conducting indium-tin oxide and copper metal. Specifically, thin-film transistors based on IGZO FN exhibit negligible performance degradation after one thousand bending cycles and exceptional room-temperature gas sensing performance. Owing to their great stretchability, these metal oxide FNs can be laminated/embedded on/into elastomers, yielding multifunctional single-sensing resistors as well as fully monolithically integrated e-skin devices. These can detect and differentiate multiple stimuli including analytes, light, strain, pressure, temperature, humidity, body movement, and respiratory functions. All of these FN-based devices exhibit excellent sensitivity, response time, and detection limits, making them promising candidates for versatile wearable electronics.

[1] Department of Chemistry and the Materials Research Center, Northwestern University, Evanston, IL 60208, USA. [2] Department of Mechanical Engineering, University of Houston, Houston, TX 77004, USA. [3] Department of Biomedical Engineering, City University of Hong Kong, Kowloon Tong, Hong Kong, China. [4] State Key Laboratory of Structural Analysis for Industrial Equipment, International Research Center for Computational Mechanics, Department of Engineering Mechanics, Dalian University of Technology, Dalian 116024, China. [5] Department of Materials Science and Engineering, Northwestern University, Evanston, IL 60208, USA. [6] School of Physics and Electronics, Henan University, Kaifeng 475004, China. [7] Flexterra Inc., Skokie, IL 60077, USA. [8] These authors contributed equally: Binghao Wang, Anish Thukral. ✉email: xingeyu@cityu.edu.hk; cyu15@uh.edu; t-marks@northwestern.edu; a-facchetti@northwestern.edu

Mechanically flexible and optionally stretchable opto-electronic devices integrated with textiles (e-textile) or conformably adaptable to the human body (e-skin) are attracting great interest to create somatosensory systems that are useful for wearable sensors, health monitoring, drug dispensing, and the study of biological function, as well as for technologies that include human–machine interfaces, soft robotics, and augmented reality[1–8]. To enable mechanical stretchability in opto-electronic devices, engineered structures involving percolative charge-transporting networks in elastomers, conductive path designs (wavy, serpentine, mesh) or architectures (wrinkles, bulking) have been employed to accommodate mechanical strain during mechanical stress[9–11]. From a materials design perspective, single crystal silicon nanomembranes, metallic (Ag, Au, Cu) nanostructures (nanoparticle, nanowire (NW), nanoflake, etc.), carbonaceous nanomaterials (carbon nanotubes, graphene, amorphous carbon, etc.), and conducting polymers [poly(3,4-ethylenedioxythiophene) polystyrene sulfonate (PEDOT:PSS), polyaniline, etc.] are promising charge-carrying candidates for wearable electronic skins and textiles[12–17]. Nevertheless, there is much room for further advances in terms of multi-stimuli sensory capabilities and sensory performance (sensitivity, response time, etc.) as well as fabrication cost[18]. For instance, the transistors and memory elements for semiconducting Si nanomembranes exhibit excellent performance and stability. Furthermore, a certain degree of stretchability can be achieved via design/fabrication of wavy and nanomesh structures[19,20]. However, high-temperature and vacuum-based production of Si nanomembrane is capital- and energy-intensive. In contrast, the large-scale development and commercialization of metal oxide (MO) semiconductors (e.g., indium-gallium-zinc oxide (IGZO), ZnO, $SnO_2$) and conductors (indium-tin oxide (ITO), FTO) is more mature for rigid displays, touch, and sensor applications and highlights the attraction of MOs over other (semi)conductor families in terms of high mobility, optical transparency, and facile solution fabrication in ambient from inexpensive, benign precursors[21–23]. However, bulk crystalline/polycrystalline MOs that can sustain maximum fracture levels of ~1% are inherently difficult to stretch to the extent required for wearable electronics. Furthermore, the synthesis/processing of MO nanostructures with aspect ratios comparable to those (>1000) of other nanomaterials is challenging, and to date only hydrothermally derived ZnO NWs been explored for stretchable e-skin electronics[24,25]. In recent years, the blow-spinning method has been used for large-scale fabrication of high-aspect-ratio metal/MO nanofibers ($TiO_2$, $ZrO_2$, Ag, ITO, etc.) for applications in environmental remediation as well as high-temperature sponges and heaters[26–28]. However, no studies have addressed the fabrication of semiconducting nanofibers, which in principal could provide broad electronic tunability in microelectronics for logic elements and sensors, not to mention as active elements in wearable and stretchable devices.

Here we report the general synthesis of several representative classes of three-dimensional (3D) MO nanofiber networks (FNs) with different coverages ($C_{FN}$, the total length of fibers per unit area), including semiconducting IGZO and copper oxide (CuO), as well as conducting ITO and copper metal (Cu) by a solution-based blow-spinning technique. Using these fibrous materials, functional thin-film transistors (TFTs) and resistor-based sensors are demonstrated. TFTs based on low-coverage ($C_{FN} = 0.15 \, \mu m^{-1}$) IGZO fibers exhibit excellent mechanical flexibility (bending radius as low as 1 mm) and high sensing selectivity to $NO_2$ gas (sensitivity of 33.6% $ppm^{-1}$). More impressively, high stretchability is achieved by mounting middle-coverage (0.5 $\mu m^{-1}$) IGZO FNs on poly[styrene-b-(ethylene-co-butylene)-b-styrene] (SEBS) substrates to achieve IGZO resistors, which are capable of sensing strain (up to 50%),

ultraviolet–visible (UV-vis) light (detectivity = $5.2 \times 10^{10}$ Jones), temperature (sensitivity = 2.2% $°C^{-1}$), and exhaled breath vapors with excellent performance. Furthermore, wearable CuO FN/SEBS ($C_{FN} = 2.0 \, \mu m^{-1}$) devices are demonstrated as pressure sensors, as well as ITO FN/SEBS ($C_{FN} = 2.0 \, \mu m^{-1}$) devices as motion readers. Finally, we demonstrate a monolithically integrated IGZO+ITO+CuO resistor platform, in the form of a patch, capable of multisensory recognition.

## Results

### Fiber fabrication and characterization.
All nanofibers/FN were fabricated by a blow-spinning technique, which, compared with electrospinning, is a more cost-effective and efficient fiber spinning procedure for polymers and polymer-based materials[29,30]. The typical fabrication consists of dissolving the metal salt(s) (100–200 mg $mL^{-1}$) in ethanol, followed by the addition of an appropriate amount of polyvinyl butyral (PVB; 100 mg $mL^{-1}$) and stirring the solution for 2 h. The use of this particular polymer is found to be critical since when combined with the MO salt precursors it enables efficient fiber formation in contrast to other polymers we examined such as polyvinylpyrrolidone and polyethylene oxide. This precursor solution is then pumped into an air brush fixture with a 0.3-mm nozzle and, simultaneously, 150 kPa air is passed into the air brush through a concentric outer nozzle (Fig. 1a). As shown in Supplementary Movie 1, the precursor solution is expelled by the air stream and solidifies on the substrate/target as inorganic–polymer hybrid fibers. This method is highly efficient and scalable, producing $8 \times 14$ $cm^2$ area PVB-based fabric in ~5 min, and is only equipment-limited (Fig. 1b and Supplementary Fig. 1). As illustrated in Fig. 1c, d, optical and scanning electron microscopy (SEM) images reveal that neat as-spun inorganic–PVB fibers have diameters of 500–900 nm with lengths of several centimeters. Finally, the fabric is annealed using material-dependent optimized conditions to thermolyze the PVB and obtain a fibrous inorganic network (see "Methods" section for details). FNs of different fiber coverages can be obtained by simply changing the deposition time, and $C_{FN}$s of ~0.15, ~0.5, and 2.0 $\mu m^{-1}$ were used here for specific demonstrations (Supplementary Fig. 2).

Figure 1e–h and Supplementary Figs. 3 and 4 show representative transmission electron microscopy (TEM) images, selected area electron diffraction (SAED) patterns, and grazing incidence X-ray diffractions (XRDs) of the IGZO, ITO, CuO, and Cu fibers. The IGZO fibers have a smooth surface with diameters of 200–300 nm and exhibit typical diffuse X-ray scattering rings, indicating amorphous character. In contrast, the ITO fibers, connected by numerous crystalline nanoparticles, have an average diameter of ~180 nm, while both the CuO and Cu fibers exhibit average diameters of ~70 nm. All of the fibers exhibit an impressive average length of ~1 cm. The SAED and XRD spectra of the CuO and Cu fibers reveal their crystalline nature. The conductivities of the semiconducting IGZO and CuO fibers are 0.004 S $cm^{-1}$ and 0.01 S $cm^{-1}$, respectively, and the conductivities of the conducting ITO and Cu fibers are $1.0 \times 10^3$ S $cm^{-1}$ and $5.9 \times 10^6$ S $cm^{-1}$, respectively (Supplementary Fig. 5), and remarkably comparable to those of the corresponding bulk materials[6,31,32]. The potential mechanical flexibility of these inorganic fibers was next examined by SEM (Fig. 1i, j and Supplementary Fig. 6), which shows that some fibers have significant curvature (radius ~ 2.5 $\mu m$) but without evidence of crack formation.

### IGZO FN-based TFTs and gas sensors.
Considering the high mechanical flexibility of the present IGZO fibers (vide infra) and potential nanoscopic charge transport confinement in TFTs, a

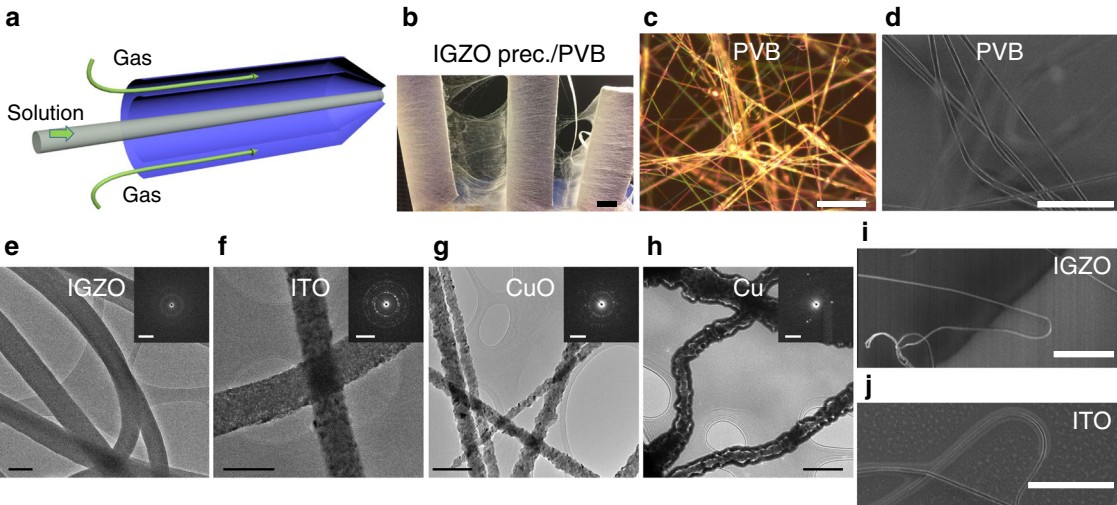

**Fig. 1 Synthesis and structural characterization of the present metal and metal oxide fibers. a** Schematic of the blow-spinning method. **b** Photograph of a metal salt/PVB fiber mat of IGZO fibers. Scale bar = 1 cm. **c** Optical image PVB fibers without metal salts. Scale bar = 50 μm. **d** SEM image of PVB fibers without metal salts. Scale bar = 10 μm. TEM images and SAED patterns of thermally processed **e** IGZO, **f** ITO, **g** CuO, and **h** Cu fibers. Scale bar = 200 nm. Insets are selected area electron diffraction patterns. Scale bar = 5 1/nm. SEM images of **i** IGZO and **j** ITO fibers in curved states with a radius of ~2.5 μm. Scale bar = 10 μm.

**Table 1 Structure, function, and representative device figure of merit reported in this study.**

| Material | Device | Device structure | Dimension | Application | Figure of merit[a] |
|---|---|---|---|---|---|
| IGZO FN | TFT ($C_{FN} = 0.15\ \mu m^{-1}$) | Al/IGZO FN/SiO$_2$/Si | $W = 1\ mm$, $L = 100\ \mu m$ | Switch | $\mu \sim 1\ cm^2\ V^{-1}\ s^{-1}$ $I_{on}/I_{off} = 4 \times 10^4$ |
| | | | | Gas sensor | RT response LOD = 20 ppb $S_G = 33.6\%\ ppm^{-1}$ |
| | Flexible TFT ($C_{FN} = 0.15\ \mu m^{-1}$) | Ion-gel/IGZO FN/Au (Ti)/PI | $W = 1\ mm$, $L = 50\ \mu m$ | Switch | $\mu \sim 0.5\ cm^2\ V^{-1}\ s^{-1}$ $I_{on}/I_{off} = 4 \times 10^4$ $r = 5\ mm$, 1000 cycles |
| | Stretchable resistor ($C_{FN} = 0.5\ \mu m^{-1}$, $R = 30\text{–}50\ M\Omega$) | PEDOT:PSS/IGZO FN/PEDOT:PSS | $W = 1\ mm$, $L = 100\ \mu m$ | Strain sensor | Gauge factor ~ 15 Elongation ≥ 50% |
| | | | | Gas sensor | RT response $S_G = 38\text{-}166\%\ ppm^{-1}$ Elongation = 50% |
| | | Cr-Au/IGZO FN/Cr-Au | | Thermistor | $S_T = 2.1\text{-}2.2\%\ ^\circ C^{-1}$ Elongation ≥ 10% strain |
| | | | | Breath analyzer | $I_{on}/I_{off} = 10^3$ Distinguish different breath frequencies and alcohol test |
| | | | | Photodetector | $R_{ph} = 16\ mA\ W^{-1}$ $I_{ph}/I_d = 403$ $D^* = 5.2 \times 10^{10}$ Elongation = 10% strain |
| | Stretchable resistor ($C_{FN} = 2.0\ \mu m^{-1}$, $R = 1.4\ G\Omega$) | Cu wire/IGZO FN/Cu wire | $1 \times 2\ cm^2$ | System integration | Differentiate solar light, temperature, stretch, and breath stimuli |
| CuO FN | Stretchable resistor ($C_{FN} = 0.5\ \mu m^{-1}$, $R = 20\text{-}30\ M\Omega$) | Cr-Au/CuO FN/Cr-Au | $W = 2\ mm$, $L = 50\ \mu m$ | Pressure sensor | $S_P = 0.04\ kPa^{-1}$ LOD = 50 Pa |
| | Stretchable resistor ($C_{FN} = 2.0\ \mu m^{-1}$, $R = 100\ M\Omega$) | Cu wire/CuO FN/Cu wire | $1 \times 2\ cm^2$ | System integration | Differentiate solar light, temperature, stretch, and breath stimuli |
| ITO FN | Stretchable resistor ($C_{FN} = 2.0\ \mu m^{-1}$, $R = 3\text{-}7\ M\Omega$) | Cu wire/ITO FN/Cu wire | $1 \times 2\ cm^2$ | Strain sensor | Gauge factor ~ 21 Elongation ≥ 50% |
| | | | | Motion recognition | Distinguish different finger gestures |
| | | | | System integration | Differentiate solar light, temperature, stretch, and breath stimuli |

[a]The values are the average of at least five samples and the maximum stdv. is <10%.

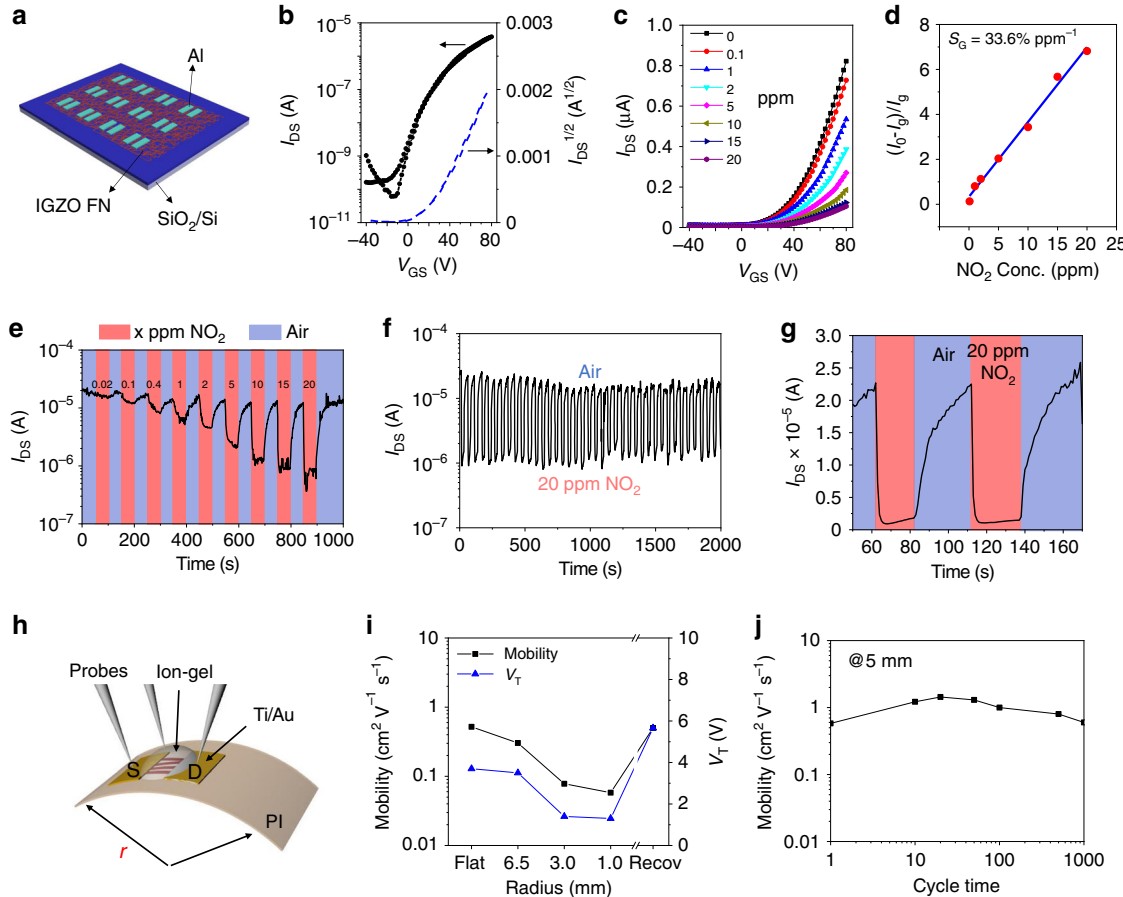

**Fig. 2 IGZO FN-based TFTs and gas sensors. a** Diagram of a rigid TFT array. **b** Typical transfer curve of an IGZO FN-based TFT. **c** Square-root of drain current vs. gate voltage plots of an IGZO fiber-based TFT at the indicated $NO_2$ concentrations. **d** Relative change of saturated current as a function of $NO_2$ concentration. **e** Real-time source–drain current ($V_D = V_G = 50$ V) response to dynamic switching between $NO_2$ concentrations (20 ppb–20 ppm). **f** Plot of source–drain current vs. time with exposure to 20 ppm $NO_2$ or air. **g** Enlarged panel in **f**, showing detailed shape of sensing curves and definition of response/recovery time. **h** Schematic representation of ion-gel gated flexible IGZO NF-based TFT on 1.5-μm-thick polyimide substrate. **i** Mobility and threshold voltage variations under the indicated bending radii. **j** Mobility variation over 1000 bending cycles at a radius of 5 mm.

series of IGZO FN-based TFTs was fabricated. Table 1 collects the IGZO FN-based TFT metrics, as well as those of other devices fabricated in this study. First, the semiconducting properties of the IGZO FNs were investigated on a rigid top-contact/bottom-gate TFT architecture fabricated by depositing IGZO FNs ($C_{FN} = 0.15 \mu m^{-1}$) on a $SiO_2/Si$ substrate, functioning as dielectric/gate contact, using the present blow-spinning method, followed by thermal evaporation of Al source/drain electrodes (~40-nm thick). Representative transfer and output plots are shown in Fig. 2b and Supplementary Fig. 7. These TFTs exhibit an average carrier mobility ($\mu_{FE}$) of ~1.0 $cm^2 V^{-1} s^{-1}$, no obvious hysteresis, a threshold voltage ($V_T$) of +20 V, and an on–off current ratio ($I_{on/off}$) of $10^4$–$10^5$. Note that the effective channel width ($W_{eff}$) depends on the fiber number/diameter, which was estimated optically. This performance is somewhat lower than that of solution-processed, dense-film IGZO TFTs, likely reflecting PVB-derived carbon contamination and charge traps on the unpassivated fiber surfaces[33,34]. Nevertheless, these FN-based TFTs have substantial semiconductor surface area-to-volume ratios, suitable for highly sensitive gas sensors. Figure 2c shows the $I_{DS}$–$V_{GS}$ curves for the IGZO TFTs exposed to different gaseous $NO_2$ concentrations at 25 °C, demonstrating significant current decreases upon gas exposure. For these experiments, $NO_2$ gas having different concentrations was employed using air (50% relative humidity (RH)) as the carrier gas to mimic the actual

environmental conditions. Since $NO_2$ is an oxidant, the adsorption of $NO_2$ molecules on the fiber surfaces act as acceptors of negative charge carriers, thus reducing the mobile electron density for the n-type IGZO channel[35,36]. The relative change of saturated current ($\Delta R/R_0 = \Delta I/I_g = |I_0 - I_g|/I_g$) vs. $NO_2$ concentration is plotted in Fig. 2d, where $I_0$ and $I_g$ denote the initial saturated current and that after exposure to the $NO_2$ (at $V_{DS} = V_{GS} = 80$ V), respectively, and $\Delta I$ is the net current change. The $\Delta I/I_g$ displays good linearity with increasing $NO_2$ concentration, and the calculated slope (sensitivity, $S_G$) is as high as 33.6% $ppm^{-1}$ for 25 °C operation. Figure 2e, f and Supplementary Fig. 8 show the dynamic sensing performance of these devices for $NO_2$ detection at concentrations ranging from 20 ppb to 20 ppm in air. After exposure to $NO_2$, $I_{on}$ decreases rapidly, and recovery occurs without any additional treatments such as vacuum degassing or thermal annealing, typically required (>300 °C) for commercial $SnO_2$ and ZnO sensors[37,38]. The large surface-to-volume ratio, the fibrous network structure, and the weak physisorption of the gas molecules to the fiber surfaces ensures fast adsorption/desorption processes. The enlarged read-out signals show current change as a function of time responding to 20 ppm $NO_2$. The response and recovery times, 2–3 and 20–22 s, respectively, substantially improved over those of a control device based on an ~10-nm-thick dense IGZO film (Supplementary Fig. 8) and commercial MO $NO_2$ detectors that require high temperatures (>300 °C) to

operate. Also, previous studies on nanostructured inorganic-based gas sensors require surface decoration and/or UV irradiation, and their sensory performance (working temperature, detection limit, sensitivity, response and recovery time) is far lower than that achieved by the present IGZO FN-based TFTs[35,36,39–43] (Supplementary Table 1). Thus there is little room left for further optimization of this platform's response/recovery times by additional tuning and investigating structure-sensing behavior and investigating optimum operating conditions. However, a room-temperature gas sensor is the best choice to minimize energy consumption.

Next, flexible IGZO FN ($C_{FN} = 0.15\,\mu m^{-1}$)-based TFTs were demonstrated on 1.5-$\mu m$ ultrathin polyimide (PI) substrates. This polymer was chosen because of its lightweight, solution processability from precursors, considerable mechanical flexibility, and thermal high stability[5]. These top-gate/top-contact devices utilize an ion-gel gate dielectric, poly(vinylidenefluoride-co-hexafluoropropylene)/1-ethyl-3-methylimidazolium bis(trifluoromethylsulfo-nyl)-imide (PVDF-HFP/[EMIM][TFSI]; $C_i = 10.7\,\mu F\,cm^{-1}$) and e-beam evaporated Ti/Au (20/100 nm) S/D electrodes (structure in Fig. 2h). The transfer curves acquired under different bending radii are shown in Supplementary Fig. 7, and the calculated mobility/threshold voltages ($V_T$) are plotted in Fig. 2i. These TFTs function at low voltage (<10 V) and exhibit an average $\mu_{FE}$ of $0.52\,cm^2\,V^{-1}\,s^{-1}$ and an $I_{on/off}$ of ~$10^4$. Upon bending the TFT to a 1-mm radius, the $\mu_{FE}$ decreases from 0.52 to $0.06\,cm^2\,V^{-1}\,s^{-1}$ and $V_T$ shifts from $+3.7\,V$ to $+1.3\,V$, but both metrics recover to the original values when bending is reversed. Also, the mobility remains constant even after 1000 bending cycles (Fig. 2j) demonstrating that the IGZO fibers are not damaged by mechanical bending.

**IGZO and CuO FN-based devices.** Before integrating MO FNs into functioning stretchable devices, the tolerance to mechanical strain was first investigated for an IGZO single fiber and the corresponding FNs ($C_{FN} = 0.5\,\mu m^{-1}$) on SEBS substrates (>600% elongation and tenacity); this was accompanied by a finite element analysis (FEA). Figure 3a, b show optical images of a single IGZO fiber on SEBS before and after application of a 5% tensional strain along the fiber axis, respectively, revealing crack formation (gap = $1.25\,\mu m$) at this strain. The crack density and gap distance increase as the strain increases (Supplementary Figs. 9 and 10). Figure 3c and Supplementary Fig. 11 show two crossed IGZO fibers and a bent IGZO fiber under large strains (40–50%). Interestingly, the fiber/part parallel to the stress direction exhibits several cracks while the perpendicular one/part do not break, indicating that fiber orientation has a significant influence on fiber structural integrity upon stretching. Thus, when considering a randomly oriented FN, those fibers not aligned with the strain direction are much less stressed, which should prevent catastrophic rupture of the conductive pathways under large strains (Supplementary Fig. 12). To understand this aspect quantitatively, FEA simulations were next carried out. Figure 3d shows the computed relationship of the angle between a fiber long axis vs. the tensional direction ($\alpha$) and strain ($\varepsilon$) exerted on the substrate to achieve a 1.1-GPa stress on the fiber, by assuming the IGZO fiber forms a crack at 0.8% strain[44,45]. Clearly $\varepsilon$ increases polynominally with $\alpha$. These data clearly demonstrate that, for achieving the maximum stress (1.1 GPa) on the fiber, a strain of only 0.8% and considerably more, of 22%, must be applied to the surrounding elastomer when $\alpha = 0°$ (Fig. 3e) and 40° (Fig. 3f), respectively.

Since Cr/Au electrodes crack upon large mechanical deformations[46], to examine the charge transport of the IGZO FN-based resistors under different mechanical strains, the highly stretchable

conducting composite PEDOT: PSS)/[EMIM][TFSI] was used as the electrical contacts ($L = 100\,\mu m$)[47] (Fig. 3g and Supplementary Fig. 13; Table 1). Note that the entire device array is transparent with an ~80% transmittance in the visible (Supplementary Fig. 14). The $I$–$V$ curves of the IGZO FNs with the PEDOT: PSS/[EMIM][TFSI] electrodes were measured under different tensional strains (0–50%, Fig. 3h and Supplementary Fig. 15), and the resulting $\Delta R/R_0$ values are plotted in Fig. 3i. No major resistivity change is observed when the IGZO FN film is stretched up to 25%, at which point $\Delta R/R_0$ increases considerably to 5.1 when stretched to 50%. Importantly, the resistivity returns to near the pristine state after recovery, thus demonstrating the superior stretchability of this IGZO FN. Note that $\Delta R/R_0$ strain simulations based on a randomly oriented IGZO FN exhibit a trend similar to that measured experimentally (Fig. 3i). Furthermore, 5000 stretching cycles (strain 0–10%) do not degrade the electrical response of these IGZO FNs (Fig. 3j).

Owing to the unusual charge transport properties and excellent stretchability of the IGZO FN/SEBS films, they were next utilized to fabricate a multifunctional electronic device for sensing light, chemicals, force, and temperature (Table 1). Since charge transport in amorphous IGZO greatly depends on the oxygen vacancy content, which can be manipulated by photo-induced electron–hole pair formation[48], the IGZO FN-based resistors were first used for stretchable, wearable monitoring of UV radiation. The electronic properties of these stretchable devices were evaluated under a 0% and 10% strain at a 365 nm wavelength and an intensity of ~$7.3\,mW\,cm^{-2}$ (Fig. 4a and Supplementary Fig. 16). The photoresponsivity ($R_{ph}$) is defined as $R_{ph} = (I_{ph} - I_d)/AL_{ph}$, where $I_{ph}$ and $I_d$ are the currents under UV light and in the dark, respectively, $A$ is the effective area of the detector, and $L_{ph}$ is the intensity of the UV light[49]. Another important parameter of a photodetector is the specific detectivity ($D^*$), which is calculated from the equation $D^* = \frac{R_{ph}}{(2qJ_d)^{1/2}}$ where $q$ is the absolute value of electron charge and $J_d$ the current density under dark[50]. The calculated $R_{ph}$, on/off current ratio ($I_{ph}/I_d$), and $D^*$ at a voltage of 5 V are shown in Fig. 4b. Note that $R_{ph}$ decreases from 36 to 16 mA $W^{-1}$ when the device is stretched to 10% strain, while $I_{ph}/I_d$ and $D^*$ increase from $123/4.4 \times 10^{10}$ Jones (no strain) to $403/5.2 \times 10^{10}$ Jones (10% strain), owing to the increased effective sensing area achieved on stretching. These metrics rival or exceed those of other MO-based photodetectors[51,52].

Next, the same IGZO FN resistor was utilized as an artificial nose to detect a poisonous gas ($NO_2$). Figure 4c show the dynamic response to gas exposure, revealing that independent of the degree of strain, this device exhibits excellent response and recovery behavior, which occurs within 5 s. The $\Delta I/I_g$ for the unstrained device is 7.6, which increases to 33.2 under a 10% strain possibly due to the enlarged sensing area, then it decreases to 9.3 under a 50% strain. At 50% strain, although the sensing area is further increased, the resistance of the IGZO FN-based devices increases greatly due to greater densities of broken fibers. Thus the effective fiber number decreases, resulting in a decreased response. Thus the $S_G$ is 38.0% $ppm^{-1}$ (no strain), 166% $ppm^{-1}$ (10% strain), and 46.5% $ppm^{-1}$ (50% strain), respectively. Also, as shown in Fig. 4d and Supplementary Fig. 17, this device is highly selective to $NO_2$ as verified by control experiments carried out in the presence of other gases—20 ppm $NH_3$, $10^4$ ppm $CO_2$, and $10^3$ ppm $H_2$. Finally, the function of the same IGZO FN sensor as a temperature sensor/thermistor was investigated over the temperature range 35–75 °C. Figure 4e and Supplementary Fig. 18 show the IGZO FN device resistance dependence (at 5.0 V) on temperature with/without a 10% strain, which clearly indicates a thermally activated electron de-trapping mechanism and a negative

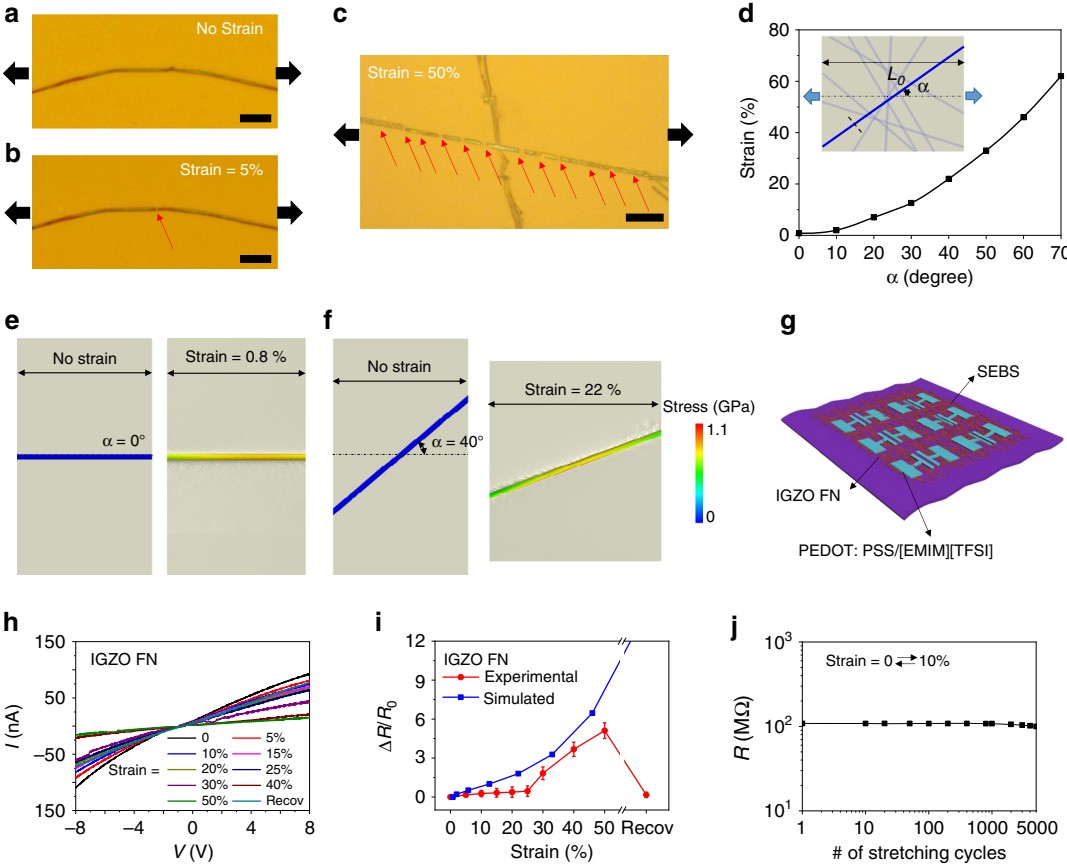

**Fig. 3 Experimental and theoretical results for IGZO single fiber and FN under strain. a**, **b** Optical images of an IGZO single fiber on SEBS with/without a 5% tensional strain; stretching directions are parallel to the fiber long axes and are indicated by black arrows. Scale bar = 10 μm. **c** Optical image of two crossed IGZO fibers on a SEBS substrate under a 50% strain. Scale bar = 10 μm. **d** Simulated strain needed to achieve 1.1 GPa stress on single IGZO fiber when fiber has different angles (α) with respect to the stretching direction. Simulated stress distribution on a single fiber when reaching fracture, i.e., maximum stress in fiber to reach fracture at 1.1 GPa. Single fiber has a **e** α = 0° and **f** α = 40° angle with respect to tensile direction. **g** Schematic of stretchable IGZO FN resistors. **h** I–V curves of IGZO FN devices under different tensile strains. **i** Experimentally derived resistance change (red) of IGZO FN devices under different strains and simulated resistance change (blue) of a randomly oriented IGZO FN under different strains. **j** Resistance of an IGZO FN device after 5000 0–10% strain cycles.

temperature coefficient[53]. The resistance exhibits a linear temperature dependence, and the resistance sensitivity ($S_T$), defined as $S_T = \delta(\Delta R/R_0)/\delta T \times 100\%$, is estimated to be 2.1% °C$^{-1}$ (no strain) and 2.2% °C$^{-1}$ (10% strain) via a linear least-squares fitting, which is significantly higher than for previously reported temperature sensors fabricated with IGZO films (0.5% °C$^{-1}$), graphene (oxide)/polymer films (1.4% °C$^{-1}$), and P3HT/polydimethylsiloxane (PDMS) films (1.6% °C$^{-1}$)[6,54–59] (Fig. 4f). Moreover, the present IGZO FN sensors also register response to RH. As shown in dynamic I–t curves (Supplementary Fig. 19), these devices exhibit good cycling response behavior, with the current increasing with increasing RH since $H_2O$ adsorption donates electrons to the oxide lattice[60]. Thus, when the IGZO FN devices are cyclically exposed to dry air (RH < 1%) and ambient air (RH = 50%), the relative responsivity $\Delta I/I_0$ is 23% and the response and recovery times are 2 s and 12 s, respectively. Higher $\Delta I/I_0$ (260%) and longer recovery times (120 s) are observed for the device cyclically exposed to 100% RH air and ambient air.

Owing to the excellent gas, temperature, and humidity sensitivities and selectivity of the present IGZO FN resistor (Supplementary Fig. 20 and Supplementary Note 1), it was also applied to human breath analysis, a non-invasive tool to diagnose/monitor respiratory diseases and volatile substance intake[61]. There are two diagnostic abnormal breathing symptoms, bradypnea and tachypnea. Bradypnea, induced by medicines,

toxins, head injuries, etc., is abnormally slow breathing, while tachypnea, induced by $O_2$ deprivation, overheating, anxiety, etc., is associated with abnormally rapid breathing. Figure 4g demonstrates that the IGZO FN sensor outputs a rapid current response (within 2 s) to exhaled breath gas and distinguishable patterns of different respiratory states. When breathing at a slow rate (~3–4 breaths min$^{-1}$), the current recovers to the initial value (~40 nA). In contrast, the current cannot fully recover and remains at a high level (300–6000 nA) for breathing at a normal rate (~15 breaths min$^{-1}$). To mimic tachypnea caused by $O_2$ deprivation, the person under testing jogged for 5 min, resulting in a respiratory rate of ~30 breaths min$^{-1}$. In response, the IGZO FN device responds with an ~8× increase in saturation current, reaching ~5 × 10$^4$ nA, which is mainly due to the elevated breath temperature and $H_2O$ vapor after exercise. The current change is only 2–3× the minimum inhaling current of ~1.6 × 10$^4$ nA during rapid breathing. In short, through monitoring current changes, IGZO FN devices can distinguish different breathing rates, elevated temperature, and humidity in exhaled gas after physical activity.

In addition, the wearable IGZO FN devices can also detect human alcohol consumption (right part of Fig. 4g). When testing at a slow breathing rate, the saturation current after drinking 200 mL beer decreases by 50% vs. that without alcohol, and the current change during inhaling and exhaling is 10–20× lower

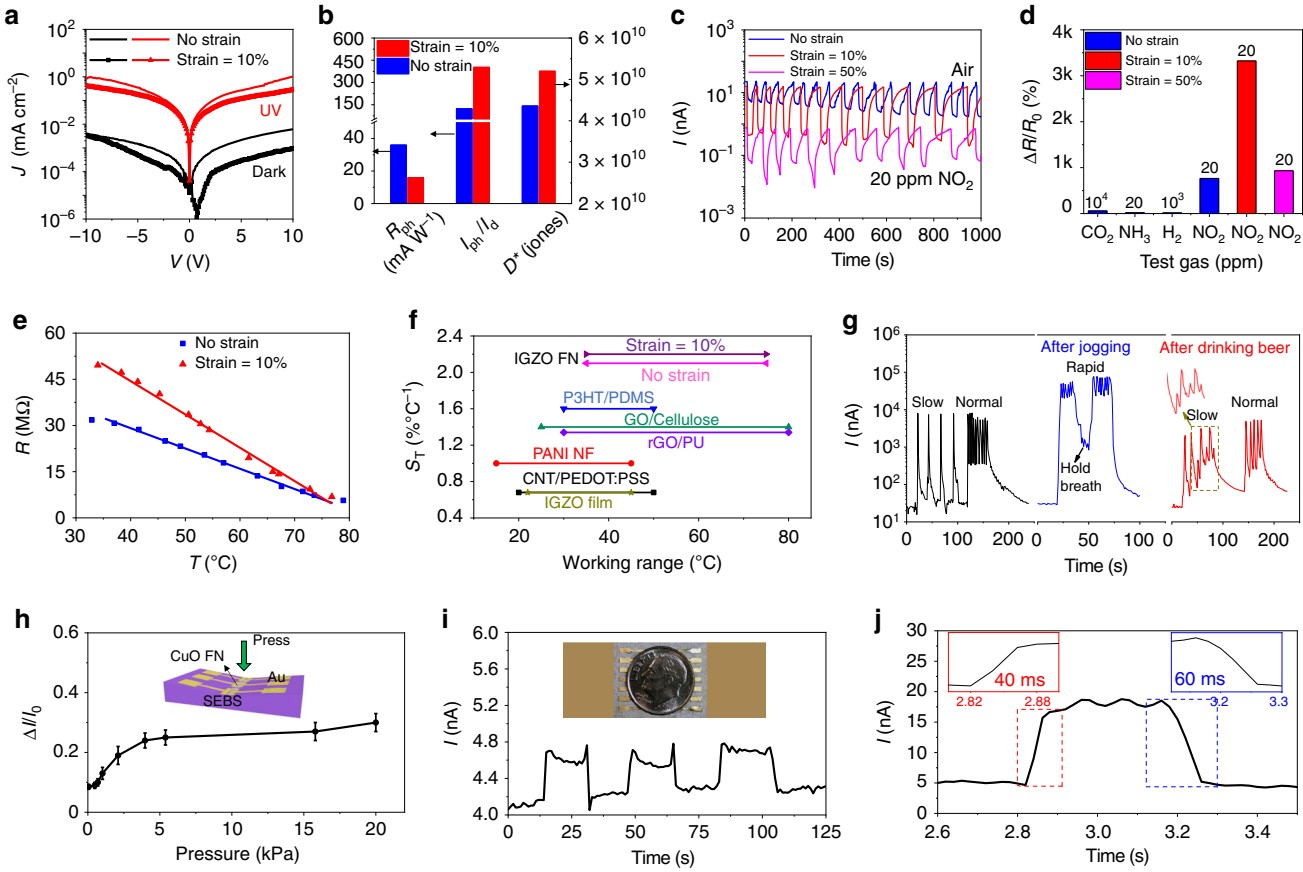

**Fig. 4 Sensing performance for different types of FN-based resistors. a** Normalized $I–V$ curves of an IGZO FN/SEBS resistor (no strain and 10% strain) in the dark and upon UV exposure. **b** Calculated photoresponsivity, on/off current ratio, and resistor detectivity under no strain and 10% strain. **c** Cycling test for an IGZO FN/SEBS sensor exposed to 20 ppm $NO_2$ under no strain, 10%, and 50% strain (bias = 5 V). **d** Relative resistance changes of IGZO FN/SEBS sensor exposed to 20 ppm $NH_3$, 20 ppm $NO_2$, 1000 ppm $H_2$, and 10,000 ppm $CO_2$. **e** Resistance change of IGZO FN/SEBS device over a 35–75 °C temperature range under no stain and 10% strain. **f** Comparison of temperature response sensitivity with literature reports. **g** Current change of a wearable resistor exposed to exhaled gas during respiration. Inset: enlarged $I–t$ curves for slow breath rate after drinking alcohol. **h** Relative current of a CuO FN/SEBS pressure sensor exposed to different pressures. Inset: schematic of a CuO FN/SEBS pressure sensor. **i** Dynamic $I–t$ curve of pressure sensor by loading and unloading a dime covered by a piece of tape. Inset: schematic of a dime on CuO FN/SEBS pressure sensor. **j** Enlarged $I–t$ curve when a finger contacts and disengages the sensor. Inset: enlarged plots show the response and recovery time.

than that without alcohol (~200×). The mechanism is likely similar to that of water due to the similarity in the chemical structures (H-O-H vs. Et-O-H), with a decrease of the resistance upon surface exposure. The reduction is lower than for water probably because the acidity of water is greater than that of ethanol. Note that small peaks appear between the inhaling/exhaling breathing cycles likely due to desorption/evaporation of the alcohol molecules. When breathing normally, such small peaks disappear in the next cycle, exhaling coming before the alcohol desorption.

Since the present CuO fibers exhibit a higher Young's modulus (200–300 GPa) than that of IGZO fibers (~110 GPa), this should endow CuO FN-based devices with a higher sensitivity to mechanical strain[6,62]. Therefore, the CuO NFs ($C_{FN} = 0.5\,\mu m^{-1}$) were employed to fabricate a resistor highly sensitive to pressure (Fig. 4h). It is expected that the contacts between the loosely packed nanofibers will increase with increasing pressure, thus increasing the conductivity[63]. The current response to different pressures (0–20 kPa) is shown in Fig. 4h and Supplementary Fig. 21, indicating that this device is sensitive to pressures < 4 kPa with a sensitivity ($S_p$) of 0.04 $kPa^{-1}$. Also, this device can sense the very small pressure exerted by a dime (pressure ~ 50 Pa) and exhibits very short response/recovery times (40/60 ms) when a finger touches/leaves the CuO FN/SEBS device (Fig. 4i, j and

Supplementary Fig. 22). Note that we attempted to fabricate TFTs using CuO FNs; however, these devices could not be switched off, which is typical of p-type oxides[64].

**Multifunctional and wearable FN-based e-skins.** Using the present blow-spinning method, free-standing IGZO, ITO, and CuO FN films ($C_{FN} \sim 2.0\,\mu m^{-1}$, size = 1 × 2 cm²) were produced that can be laminated directly onto SEBS substrates and contacted with Cu leads (Fig. 5a and Supplementary Fig. 23, Table 1). Owing to the greater conductivity of ITO vs. IGZO and CuO FN films (see Table 1), ITO FN/SEBS devices were used for gesture recognition, in which large strains (up to 20%) occur near one's hand joints[13]. The resistance of the ITO FN film at rest is 7.7 MΩ, and the relative resistance change was measured for strains from 0% to 50% (Fig. 5b). The resistance of the ITO FN/SEBS device increases gradually with strain but returns to the initial value after releasing the strain. The mechanism of resistance change upon strain is similar to that mentioned for the IGZO FN-based devices above. Also, this device is quite stable with similar $\Delta R/R_0$ strain behavior even after storage in ambient for 1 year (Fig. 5b). Next, the resistance change of this platform under different degrees of finger movements, which correspond to different extents of bending angle/strain, was evaluated. The electrical resistance is 7.7, 12.0, 16.0, and 18.0 MΩ for bending angles near 0° (no

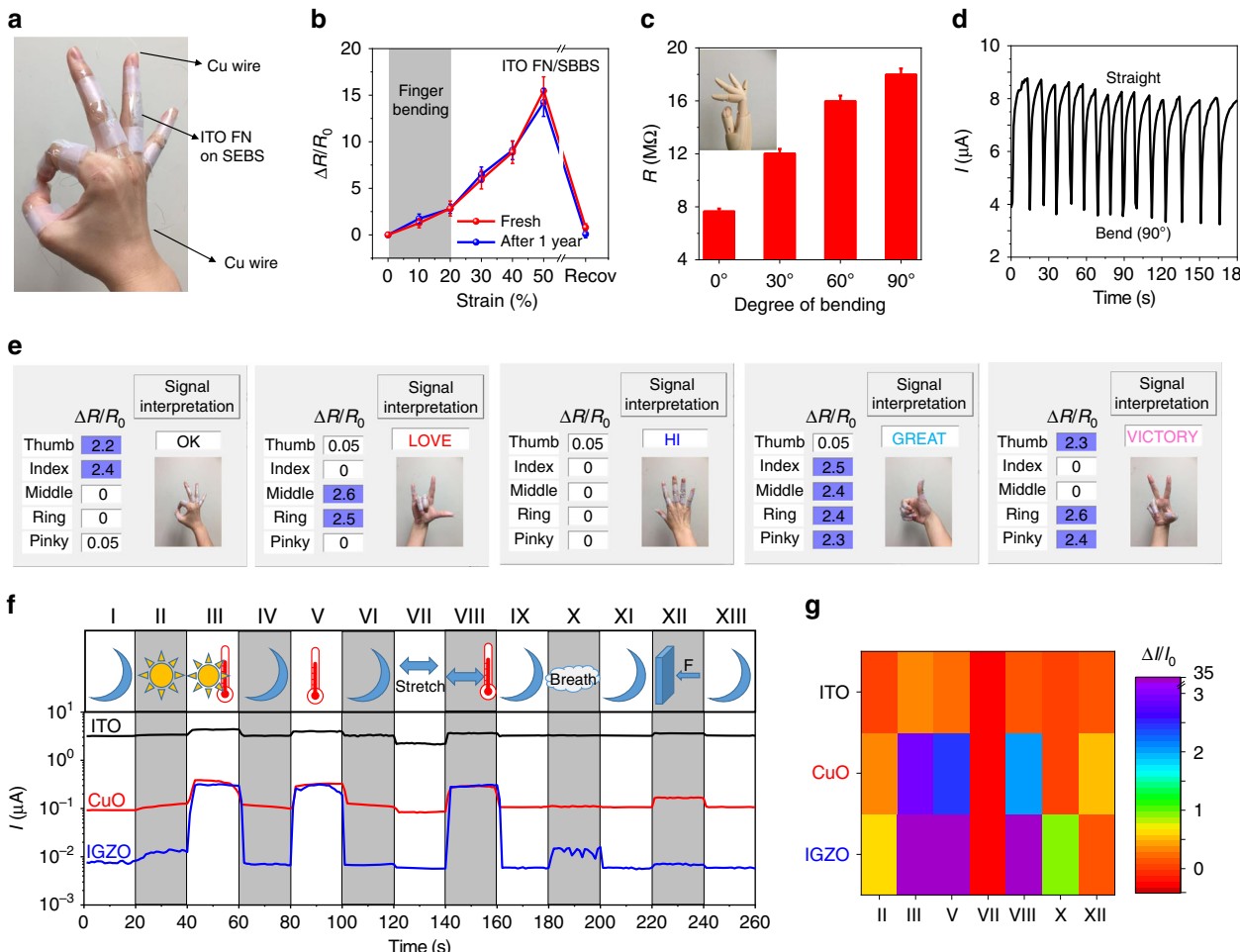

**Fig. 5 Wearable ITO FN-based devices and monolithically integrated devices for sensing and differentiating stimuli. a** Photograph of a hand fitted with ITO FN-based sensing devices. **b** Resistivity change of fresh and 1-year-storage ITO FN-based device under different strains. The gray region indicates the strain range when bending fingers. **c** Electrical resistance of ITO FN-based devices under different degrees of finger bending. **d** Cycling test for wearable ITO FN-based devices bending from 0° to 90°. **e** Demonstration of an array of strain sensors placed on a hand to translate gestures. Box is in light blue when the change of electrical resistance is >2. **f** Real-time simultaneous sensing of ambient solar light (AM 1.5 G illumination), temperature (~40 °C), stretching (~10% strain), exhaled gas, and pressure (10 kPa), with an integrated IGZO, CuO, and ITO e-skin. **g** Relative current change of IGZO, CuO, and ITO resistors under different stimuli.

strain), 30° ($\varepsilon \sim 8\%$), 60° ($\varepsilon \sim 13\%$), and 90° ($\varepsilon \sim 15\%$), respectively (Fig. 5c and Supplementary Fig. 24). The cycling tests in Fig. 5d and Supplementary Movie 2 demonstrate the excellent operational stability under hand motion, and response/recovery times are as short as 2/7 s. Next, the direct correlation between the $\Delta R/R_0$ of the strain sensor with specific gestures was monitored by mounting five strain sensors on the finger joints of a hand. For example, to express the sign of "OK", the thumb and index fingers are completely folded (providing a $\Delta R/R_0$, >2.0), whereas the middle, ring, and "pinky" fingers are straightened (showing a $\Delta R/R_0$, <0.2). Figure 5e shows a set of gestures, representing "OK," "LOVE," "HI," "GREAT," and "VICTORY," which was translated by configuring the hand into specific corresponding gestures.

Finally, a fully integrated wearable e-skin device comprising self-standing IGZO, CuO, and ITO FNs monolithically integrated on the same platform was laminated onto the hand of a person in the form of a band patch (Supplementary Fig. 25). Figure 5f, j show real-time signal recording of different stimuli, solar light exposure, temperature, strain, and exhaled gas. The procedure involves the subject person standing in an air-conditioned laboratory office (~22 °C) (I), exiting the office into sunlight

(II), walking in sunlight for several minutes (III), entering an air-conditioned coffee shop (IV), placing a hot cup of coffee near the sensor (V), then removing it (VI), bending the hand (VII), and then placing the sensor near a hot cup of coffee (VIII), leaving the table and extending the hand (IX), placing it near the mouth and exhaling on it five times (X), and next removing it (XI), then pressing the hand on a table (XII), and finally positioning it at rest (XIII). From these data and the previous discussion, the IGZO FN sensor can efficiently differentiate solar light, temperature, stretching, and breath stimuli. Specifically, solar light has slight positive effect on the conductivity, while increasing temperature increases it substantially. Breath leads to typical cycling changes while stretching decreases the conductivity. However, this IGZO FN device alone cannot distinguish temperature-involved stimuli (activities III, V, and VIII), while these are effectively detected by the CuO and ITO sensors (see $\Delta I/I_0$ variations in Fig. 5g). Regarding differentiating among stimuli II and X (or among V, VII, and XII), while the ITO device cannot achieve that, the CuO and IGZO sensors perform well. In short, the integrated MO NW-based e-skin platform enables a human–machine interface that differentiates among multiple human stimuli with high recognition levels.

## Discussion

In this study, we demonstrate the efficient production at scale of several MO and metal nanofibers and FNs using a simple blow-spinning technique employing formulations of metal salt(s) and a polymer. The MO FNs were integrated with elastomeric substrates to achieve a new class of FN-based semiconductor and conductor platforms for inorganic electronics and sensors that are highly stretchable without the need for additional mechanical structure designs required for traditional nonstretchable inorganic materials. The enhanced FN film mechanical properties can be understood by computational and experimental analyses showing the capacity of these systems to sustain greater strain reflects the diverse fiber directionalities in the 3D FNs. The present results show that flexible transistors remain functional when bent with a radius of 5 mm for 1000 times and that stretchable resistors function well after 50% stretching and long-time storage. Wearable sensors operate in a very stable manner while achieving remarkable performance in sensing various combinations of strain, gas, light, temperature, pressure, and exhaled gases. The present monolithically integrated MO NW-based sensor system enables human–machine interactions with multiple sensing functionalities and high levels of recognition. We foresee that this new strategy of fabricating elastomeric inorganic semiconductors/conductors will advance the development of very stable and sensitive stretchable electronic functionality. Furthermore, the integration demonstrated will promote advances in stretchable electronics for a wide range of applications, such as artificial skins, biomedical implants, and surgical gloves.

## Methods

**Materials and fiber precursor solution preparation.** Metal salts, PVB, ethanol, the ionic liquid ([EMIM][TFSI]), and PVDF-HFP were purchased from MilliporeSigma; PEDOT:PSS solution (Clevios H1000) was purchased from Heraeus Group; and SEBS polymer (G1643 MS) was kindly provided by Kraton corporation. PDMS (SYLGARD™ 184) were purchased from Dow Corning. PI precursor (PI-2545) was purchased from HD MicroSystem. n++-Silicon wafers with 300 nm SiO₂ were purchased from WRS Materials, Cu wire (0.1 mm diameter) was purchased from Alfa Aesar, and woven wire mesh (stainless steel, 18 mesh) was purchased from Amazon.com. For precursor solution preparation, the requisite amount of metal salt(s) [354.13 mg of $In(NO_3)_3$ and 45.87 mg $SnCl_4 \cdot 5H_2O$ for ITO; 267.89 mg of $In(NO_3)_3$, 56.19 mg of $Zn(NO_3)_2$, and 75.93 mg of $Ga(NO_3)_3$ for IGZO; 400 mg $Cu(NO_3)_2$ for Cu/CuO] was dissolved in 2 mL of ethanol. Next 200 mg of PVB was added in the metal salt(s) solution and it was stirred for ~1 h before dispensing.

**Fiber fabrication.** n++-Silicon wafers with 300 nm SiO₂ were cleaned with acetone and isopropyl alcohol and then treated with an oxygen plasma for 5 min before use. The precursor solution was loaded into the spray gun with a 0.3-mm nozzle and sprayed on the substrate/aluminum foil placed at a distance of 20 cm using a flow rate of 0.1 mL min⁻¹ and a gas pressure is 100 kPa. The PVB precursor fibers were collected on SiO₂/Si substrates or aluminum foils, then the salts/PVB FNs were annealed at optimized temperature and atmosphere. Specifically, the IGZO precursor/PVB fibers were annealed in a furnace heated from room temperature (RT) to 450 °C in air at a heating rate of 10 °C min⁻¹ and then held at 450 °C in air for 15–30 min followed by naturally cooling down in the furnace (refer to the DSC/TGA results in Supplementary Fig. 26). The ITO precursor/PVB fibers were annealed from RT to 500 °C in air at a heating rate of 10 °C min⁻¹ and then held at 500 °C in air for 1 h before being cooled down in the furnace. To increase the conductivity, the resulting ITO fibers were further annealed from RT to 300 °C in 5%H₂/N₂ at a heating rate of 2 °C min⁻¹ and then held at 300 °C in air for 1 h before being cooled down. The CuO precursor/PVB fibers were annealed from RT to 450 °C in air at a heating rate of 10 °C min⁻¹ and then held at 450 °C in air for 1 h before being cooled down in the furnace. To fabricate Cu fibers, the above CuO fibers were further annealed from RT to 300 °C in 5%H₂/N₂ at a heating rate of 2 °C min⁻¹ and then held at 300 °C in air for 1 h before being cooled down. The different fiber densities ($C_{FN}$ = 0.15, 0.5, and 2.0 μm⁻¹) were achieved by controlling the blow-spun time (30 s, 90 s, and 5 min, respectively).

**Fiber characterization.** Optical images were collected by a Leica optical microscope. XRD spectra were acquired with a Rigaku Smartlab workstation using CuKα (1.54 Å) radiation. Thermogravimetric analysis was performed on a SDT Q600 instrument (TA Instruments, Inc.). Experiments were carried out on ~2 mg fiber samples and the heating rate of 10 °C min⁻¹ under a 70 mL min⁻¹ air flow. TEM and cross-sectional TEM images were collected using a JEOL ARM300F transmission electron microscope. Samples for top-view TEM images were obtained by placing the fibers on water and transferring them onto TEM grids.

**Fabrication of IGZO FN-based TFTs.** Rigid IGZO FN ($C_{FN}$ = 0.15 μm⁻¹)-based TFTs were fabricated by depositing the IGZO FN directly on 300 nm SiO₂/Si substrates followed by thermal evaporation (3 × 10⁻⁶ Torr) of top source and drain electrodes (Al, 50 nm) through a metal shadow mask. The channel length (L) and nominal channel width (W) for all devices were 100 and 1000 μm, respectively. The effective W dependence on the fiber number/diameter was estimated optically for each device. Flexible IGZO FN ($C_{FN}$ = 0.15 μm⁻¹)-based TFTs were fabricated on ultrathin (~1.5-μm thick) PI substrates prepared by spin coating PI precursor solution at 3000 rpm for 60 s on a glass substrate. The coated PI thin film was partially cured at 110 °C, then IGZO FN were contact transferred onto the aforementioned partially cured PI thin film[65]. Finally, the PI film with the IGZO FN (without PDMS) was fully cured at 250 °C for 1 h. The ion-gel dielectric was prepared by dissolving 0.4 g of PVDF-HFP in 3.6 mL of acetone and then adding 1.0 mL of EMIM-TFSI ion liquid into the polymer solution. This solution was stirred at 60 °C for 6 h inside a glove box. A free-standing ion-gel film was then prepared by casting the solution on a glass slide and baking it in a vacuum oven at 85 °C overnight (areal capacitance = 10.7 μF cm⁻² at 1 Hz). The top-gate top-contact IGZO TFT was fabricated by deposition of Ti/Au (20/100 nm thick) electrodes (nominal W/L = 1000 μm/100 μm) on top of the IGZO FN by e-beam evaporation followed by lamination of the ion-gel dielectric film. A tungsten metal tip inserted into the ion-gel served as the gate contact probe. The control IGZO TFTs based on solution processed IGZO thin films (10 nm thick), used for sensing NO₂, were fabricated following a previously published procedure[66]. Specially, the IGZO precursors were spin-coated on the 300 nm SiO₂/Si substrates at 3500 rpm for 30 s in a controlled atmosphere box (RH ~ 20%) and optionally pre-annealed at 120 °C for 60 s (RH ~ 35%). Then the resulting films were immediately placed on a 300 °C hotplate and annealed for 1 min (RH ~ 35%). This process was repeated for four times to obtain the desired film thickness. Al source and drain electrodes were then deposited by thermal evaporation.

**Fabrication of stretchable resistors.** SEBS substrates (~1-mm thick) were fabricated by drop-casting an ~1.5 mL SEBS toluene solution (200 mg mL⁻¹) on a glass slide (2.5 × 7.5 cm²) and drying at RT overnight. A 25-μm-thick PI film was partially cut to define electrode patterns using a laser cutter (LPKF ProtoLaser R)[54]. The gap between two neighbor rectangular patterns was 1000 μm in width and 100 μm in length. Then the resulting patterned PI film was placed on a SEBS substrate, followed by spin-coating a PEDOT:PSS+[EMIM][TFSI] solution (obtained by dissolving 20 mg of [EMIM][TFSI] in 2 mL of PEDOT:PSS) at 1500 rpm for 1 min as stretchable electrodes. After that, the IGZO FNs ($C_{FN}$ = 0.5 μm⁻¹) were contact transferred on the above substrate, which were used for the stretchable IGZO FN strain and gas sensors. Regarding stretchable IGZO resistors for temperature, UV, and breath monitoring and the stretchable CuO pressure sensors, thermally evaporated Cr (3 nm)/Au (50 nm) were used as electrodes after transferring the IGZO FNs ($C_{FN}$ = 0.5 μm⁻¹) and CuO FNs ($C_{FN}$ = 0.5 μm⁻¹) from the sacrificial SiO₂/Si substrates onto the SEBS substrates. The channel length (L) and nominal channel width (W) for all IGZO FN devices were 100 and 1000 μm, respectively. The channel length (L) and effective channel width (W) for CuO FN devices were 50 and 100 μm, respectively.

For the stretchable ITO resistors and systematic integration of IGZO, CuO, and ITO devices, high coverage ($C_{FN}$ = 2.0 μm⁻¹) self-standing IGZO, CuO, and ITO FN films were used. First, ~1.5 mL SEBS toluene solution (200 mg mL⁻¹) was drop-cast on a glass slide (2.5 × 7.5 cm²). After 10 min, an appropriate size (e.g., 1 × 2 cm²) of self-standing IGZO, CuO, and ITO FNs were placed directly on the un-solidified SEBS substrate with the aid of a tweezer, followed by positioning two Cu wires (diameter = 0.1 mm) on the FN as external electrodes at a distance of 1.5 cm. During SEBS solidification (store in ambient overnight), the FN and Cu wires penetrate into the SEBS substrates, affording a robust structure.

**Electrical characterization.** Electrical characterizations were performed with an Agilent 4155C semiconductor parameter analyzer in ambient (RH = 30–40%). For flexible IGZO FN TFTs, the output curve was measured at a gate voltage sweep from 0 to +10 V and drain source voltage from 0 to +1 V. To characterize the transfer curves, each gate voltage sweeps from 0 to +10 V was applied with a source–drain voltage bias of +1 V. The device was mechanically deformed during electrical characterization by laminating the device on different cylindrical objects having a different radius of curvature. For strain tests, the devices were stretched by a custom-made stretcher while measuring their electrical resistance. For gas sensing tests, the IGZO FN devices were stored in an airtight test chamber. To mimic the actual atmosphere, the carrying gas of dry air was purged through saturated aqueous solution of $Mg(NO_3)_2$ to form wet air with RH of ~50%. Then a mixture of wet air and certain gas analyte (2 ppm NO₂/N₂ gas, 100 ppm NO₂/N₂ gas, 100 ppm NH₃/N₂ gas, 4% H₂/N₂, or CO₂ gas) in appropriate concentrations was introduced into the test chamber by mass flow controllers. The total flow rate for the test was ~550 sccm.

For the UV photodetector test, UV light having an intensity of ~7.3 mW cm$^{-2}$ and a wavelength of 365 nm was used to characterize the photodetector response of the devices. The dark current was measured by keeping the device in a black box to simulate a dark ambient. A voltage sweeps from $-10$ to $+10$ V was used to check the resistance of device. Further dynamic response of photodetector is observed by switching the UV lamp turning on and off and recording the time response of rise and decay of the photo-generated current in device. The effective illumination area is calculated by multiplying fiber number and diameter. The mechanical deformation was provided by uniaxial stretching of the device. For temperature sensing, a resistive heater was used to increase the temperature environment, which was monitored using a Fluke 572-2 high-temperature infrared thermometer in direct contact with the devices. The current variations of the devices were measured at a fixed voltage bias of 5 V. The mechanical deformation was provided by uniaxially stretching the device. For the wearable breathing gas test, the devices were placed close to the nose while connecting them to the Agilent B1500A semiconductor parameter analyzer. Different breathing frequencies (slow, normal, fast) under different conditions (normal, after jogging, or drinking beer) were recorded.

For humidity sensing, the IGZO FN/SEBS device was exposed to air with three different humidity (1%, 54%, 100%). The current variations of the devices were measured at a fixed voltage bias of 10 V. For pressure sensing, pressure was applied to the sensing area with pressure from 50 Pa to 20 kPa. To test the response and recovery times, a B1500A semiconductor analyzer was employed using the $I$–$t$ set-up supplied with the software. We measured the current changes with a sampling interval of 20 ms at a given voltage when the index finger touched/left the CuO FN/SEBS device quickly. The finger was covered by a thick tape to eliminate thermal transport.

Regarding the monolithically integrated platform, the stretchable IGZO, CuO, and ITO FN devices were laminated on the hand of person in the form of a band patch. An Xe arc lamp of a Spectra-Nova 300 W Class-A solar simulator was used to simulate solar light, which has an AM 1.5 G irradiation (100 mW cm$^{-2}$). Approaching a hot coffee can elevate the temperature around device platform. Bending the finger leads to an ~10% strain and holding the cup of coffee induces strain, pressure, and temperature elevation. Breath analysis is achieved by exhaling gas on the device platform and pressure (~10 kPa) is applied by pressing the table. For the $I$–$t$ curve of the monolithically integrated platform, baseline corrections were made using the Origin software.

**Finite element analysis**. The commercial software ABAQUS was utilized to analyze the deformation of FNs embedded in an elastomeric film and to predict the relationship between the tensile strain of the film and the fracture of the fiber with different $\alpha$. 3D solid elements (C3D10M) were chosen for the heterogeneous elastomeric substrate and fibers. A convergence test of the mesh size was performed to ensure accuracy, where the minimal element size was 1/20 of the diameter of the fiber. The elastic modulus ($E$) and Poisson's ratio ($v$) of inorganic fibers are 110 GPa and 0.34, respectively[44,45]. The crack strain is taken as 0.8%. The elastic modulus ($E$) and Poisson's ratio ($v$) of substrate are 4.8 MPa and 0.5, respectively. A simplified theoretical model is established to qualitatively analyze the relationship between the resistance change and tensile strain of the film. The model assumes that there are $n$ straight fibers evenly embedded in the film as shown in Fig. 3f, where $\alpha$ is the angle between the fiber and the stretching direction, and the mutual interference between fibers is ignored. The total resistance of the film can be expressed as $R_0 = R^*/2n$, where $R^*$ is the resistance of a single fiber with length $L_0$. If the fibers have $\alpha < \alpha^*$ fracture, the ratio of the resistance change can be expressed as $(R - R_0)/R_0 = \sin\alpha^*/(1 - \sin\alpha^*)$. In addition, the FEA results show that the fiber with a larger $\alpha$ can withstand larger tensile strain due to the rotation of the fiber, i.e., the fibers with smaller $\alpha$ will fracture first under stretching.

## Data availability
The authors declare that the all the data supporting the finding of this study are available within this article and its Supplementary Information files and are available from the corresponding author on reasonable request.

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

## Acknowledgements

We thank AFOSR (FA9550-18-1-0331), the Northwestern U. MRSEC (NSF DMR-1720139), NSF (CMMI-1554499, EFMA-1935291, CNS-1931893), City University of Hong Kong (9610423), and Flexterra Corp. for support of this research. This work made use of the J. B. Cohen X-Ray Diffraction Facility, EPIC facility, Keck-II facility, and SPID facility of the NUANCE Center at Northwestern U., which received support from the Soft and Hybrid Nanotechnology Experimental (SHyNE) Resource (NSF NNCI-1542205); the MRSEC program (NSF DMR-1121262) at the Materials Research Center; the International Institute for Nanotechnology (IIN); the Keck Foundation; and the State of Illinois, through the IIN.

## Author contributions

B.W., T.J.M., and A.F. conceived and designed the research project. B.W., A.T., L.L., and X.Z. carried out the experiments. Z.X. and X.Y. conducted the simulation work. W.H. and C.Y. helped analyze the data. B.W., T.J.M., and A.F. wrote the paper. All authors discussed, revised, and approved the manuscript.

## Competing interests

The authors declare no competing interests.
