## [Peer Review File · Nature Communications]

Reviewers' comments:

Reviewer #1 (Remarks to the Author):

The authors have presented an incredibly study on inorganic nanofibers for stretchable multimodal physical and chemical sensors. The work utilized a scalable newly-developed blow-spinning technique for synthesis of several semiconducting and conducting metal oxides/metal nanofibers. These fibers creatively combined with soft substrates, exhibit good stretchability and transistor performance. More importantly, these fibers-based devices exhibit unprecedented sensory performance with regarding to gas, light, temperature, breath gas, and pressure. In order to differentiate these sensing objects, the authors also demonstrated that integrated devices based on three different oxide nanofibers which show high recognition. Therefore, based on these results, these rarely-reported oxide fibers-based systems can be competing candidates for stretchable sensor devices. Overall, the quality of the work and the manuscript are very high, and will undoubtedly be of interest to a broad audience. I would recommend it to be published after minor revision.

1. Some recent papers regarding to electro-spun/draw-spinning metal oxides fibers on rigid substrates should be cited and compared.
2. Generally, metal oxide thin film is fragile under cycling bending test. Is there any difference in fiber format? Please comment this issue.
3. The mobility extraction for the IGZO fiber FET should be mentioned more clearly. The mobility will be underestimated if the parallel pate mode (often used for thin film transistor).
4. More detailed explanations on the excellent recovery ability for NO₂ sensing should be given.
5. Is there any filed effect was observed for CuO fiber? This may be interesting for the construction of complementary electronics.

Reviewer #2 (Remarks to the Author):

In this manuscript, Wang and co-authors described a general synthesis method of 3D-inorganic metal oxide (MO) nanofiber networks (FNs), such that fibrous materials are used to demonstrate functional thin-film transistors and resistor-based sensors. The authors used a solution-based blow-spinning technique to synthesize semiconducting indium-gallium-zinc oxide and copper oxide, as well as conducting indium-tin-oxide and copper metal for nanofiber networks. When these metal oxide nanofiber networks are laminated/embedded on/into the elastomers, the monolithically integrated e-skin devices capable of the multisensory recognition could be successfully fabricated by using them. It is a meaningful contribution to the field of soft electronics, and the reviewer recommends publication of this manuscript in Nature Communications after proper revision. The following comments can be helpful for authors to improve the current work.

Comment #1: The authors presented the various classes of devices based on the 3D metal-oxide nanofiber network with different coverage (CFN). It is recommended to describe the definition of coverage (CFN) and the effect of the coverage on the device performance for specific applications.

Comment #2: The authors stated that 'Figure 2c shows the $(IDS)^{1/2} \sim VGS$ curves for these IGZO TFTs exposed to different gaseous NO₂ concentrations at 25°C. However, in Figure 2c, the y-axis

indicates IDS (mA). It is recommended to check the Figure 2c.

Comment #3: The authors stated that "The calculated R_{ph}, on/off current ratio (I_{ph}/I_d), D* at a voltage of 5V, and response time are shown in Figure 4c". However, the response time of IGZO photodetector is not described in Figure 4c. It is recommended to check Figure 4c.

Comment #4: The authors claimed that "The typical fabrication consists of dissolving the metal salt(s) (100-200 mg/mL) in ethanol, followed by the addition of an appropriate amount of polyvinyl butyral (PVB; 100 mg/mL) and stirring the solution for 2 h. The use of this particular polymer is found to be critical". But the reviewer cannot find sufficient information that explains the reason why the use of PVB is critical. The reviewer recommends that the authors to add more explanation on such an argument.

Comment #5: The authors claimed that the conductivities of the IGZO, CuO, ITO, and Cu based devices are remarkably comparable to those of the corresponding bulk materials.

- In Figure S5 figure caption, notations (a)-(d) do not match with corresponding above images.
- Providing detailed conductivities and corresponding references that the authors used to compare proposed devices with the bulk materials would be helpful.
- Some additional statements regarding the comparison with ultrathin single crystal silicon based devices including wearable transistors, circuits, memory devices would be helpful.

Comment #6: In Figure 2, the authors fabricated TFTs and analyzed their characteristics. However, in case of the integrated wearable device in Figure 4 and Figure 5, the reviewer is curious why the authors used a resistor-type device instead of a TFT-type, in which it can provide better performances in terms of the sensitivity.

Reviewer #3 (Remarks to the Author):

Comments for Nature Communications

This work proposed the efficient production at scale of several metal oxide and metal nanofibers and fiber networks (FNs) using a simple blow spinning technique employing formulations of metal salt(s) and a polymer, which shows the novelty and inspiration in this field. However, the following questions need to be addressed item by item so as to give further acceptance considerations. (Major Revision)

- (1) Why did you respectively choose PI and SEBS as the substrate for IGZO FN-based devices ?
- (2) Please point out the differences between PEDOT: PSS / [EMIM] [TFSI] and Cr / Au as IGZO FN-based resistor electrodes, do not only mention flexibility.
- (3) Please explain how IGZO FN-based resistor distinguishes strain, light, temperature, gas and humidity.
- (4) Please explain in detail the detection mechanism of IGZO FN-based TFT and IGZO FN-based resistor for gas.
- (5) What is the mechanism of IGZO FN-based resistor to detect temperature and humidity.
- (6) Please explain why the response of IGZO FN-based resistor to gas increases first and then decreases with the increase of strain.
- (7) What is the reason for IGZO FN-based devices can detect human alcohol consumption.
- (8) In addition to detecting mechanical strain, whether CuO FN-based devices are also affected by light and temperature. How to distinguish them?
- (9) Please explain why the resistance of ITO FN-based device increases with the increase of strain.
- (10) As regard to the IGZO FN-Based Thin-Film Transistors and Gas Sensors section, you should supplement the following experiments:

- (a) What is the selectivity of the gas sensor toward different reducing gases and different VOCs?
 - (b) How to exclude the influence of different humidity on the performance of gas sensors?
 - (c) How to explore the best operating temperature and the best response/recovery time?
 - (d) What's the reproductivity of the gas sensor from batch to batch, not for different device in same unit?
- (11) 10% strain interval is not strict, you should show experimental data of 5% strain interval.
 - (12) Only 1000 strain cycles are not sufficient.
 - (13) You mention that the device exhibits very short response/recovery times (40/60 ms) when a finger touches/leaves the CuO FN/SEBS device, How do you determine this time and what equipment do you use for testing, you need to reflect more detailed information in Figure 4-I.
 - (14) Some experimental images need to be re-polished in order to improve the quality of the paper.

Reviewer: 1

Comments:

The authors have presented an incredibly study on inorganic nanofibers for stretchable multimodal physical and chemical sensors. The work utilized a scalable newly-developed blow-spinning technique for synthesis of several semiconducting and conducting metal oxides/metal nanofibers. These fibers creatively combined with soft substrates, exhibit good stretchability and transistor performance. More importantly, these fibers-based devices exhibit unprecedented sensory performance with regarding to gas, light, temperature, breath gas, and pressure. In order to differentiate these sensing objects, the authors also demonstrated that integrated devices based on three different oxide nanofibers which show high recognition. Therefore, based on these results, these rarely-reported oxide fibers-based systems can be competing candidates for stretchable sensor devices. Overall, the quality of the work and the manuscript are very high, and will undoubtedly be of interest to a broad audience. I would recommend it to be published after minor revision.

We appreciate the positive comments of this Reviewer regarding our work.

1. Some recent papers regarding to electro-spun/draw-spinning metal oxides fibers on rigid substrates should be cited and compared.

Response: Thank Reviewer 1 for this suggestion. In revision we have now cite two recent relevant papers. (new Refs 23 and 30 on Page 22).

2. Generally, metal oxide thin film is fragile under cycling bending test. Is there any difference in fiber format? Please comment this issue.

Response: Excellent question! Metal oxide films are indeed fragile and cannot withstand repeated bending tests, especially for small bending radii (< 5 mm) (see refs. *Proc. Natl. Acad. Sci. USA* 2019, 116: 9230-9238). However, the fiber flexibility is greatly enhanced versus films because of its 1D structure as compared to planar structure of thin film, which can be clearly seen from the TEM and SEM images of the samples bended to radii < 10 μm (Figures 1 and S6). Furthermore, for fiber networks, intertwining the fibers further increases the stretchability as shown in manuscript Figure 3.

3. The mobility extraction for the IGZO fiber FET should be mentioned more clearly. The mobility will be underestimated if the parallel pate mode (often used for thin film transistor).

Response: We thank Reviewer 1 for this comment. In the original submission we explained this point in the Experimental Section, in the paragraph *Fabrication of IGZO FN-based TFTs* in Page 19. The effective W dependence on the fiber number/diameter was estimated optically for each device, thus, the average mobility we reported is accurate. In revision, we now state this more clearly when we report the mobility values on page 7.

4. More detailed explanations on the excellent recovery ability for NO_2 sensing should be given.

Response: This is an excellent point. We believe the excellent recovery ability of our FN-based IGZO devices is due to the surface: volume ratio, the fibronic network structure, and the weak inter-fiber bonding, which ensure rapid gaseous analyte penetration/desorption and a large surface: volume ratio for greater gaseous analyte adsorption. In revision, we now explain this in detail on Page 8.

5. Is there any field effect was observed for CuO fiber? This may be interesting for the construction of complementary electronics.

Response: This is an important question. We indeed attempted several times to switch CuO, however the results were modest at best, which is typical for p-type oxides. In revision, a comment is now made about this observation on page 15.

Reviewer: 2

Comments:

In this manuscript, Wang and co-authors described a general synthesis method of 3D-inorganic metal oxide (MO) nanofiber networks (FNs), such that fibrous materials are used to demonstrate functional thin-film transistors and resistor-based sensors. The authors used a solution-based blow-spinning technique to synthesize semiconducting indium-gallium-zinc oxide and copper oxide, as well as conducting indium-tin-oxide and copper metal for nanofiber networks. When these metal oxide nanofiber networks are laminated/embedded on/into the elastomers, the monolithically integrated e-skin devices capable of the multisensory recognition could be successfully fabricated by using them. It is a meaningful contribution to the field of soft electronics, and the reviewer recommends publication of this manuscript in Nature Communications after proper revision. The following comments can be helpful for authors to improve the current work.

We appreciate the highly positive feedback by this Reviewer and the comments to strengthen our work.

1: The authors presented the various classes of devices based on the 3D metal-oxide nanofiber network with different coverage (C_{FN}). It is recommended to describe the definition of coverage (C_{FN}) and the effect of the coverage on the device performance for specific applications.

Response: Excellent suggestions. Here the definition of coverage is the total length of fibers per unit area, which is in common use by the community (see *ACS Appl. Mater. Interfaces* 2017, 9: 10805). In revision the definition is now provided on Page 3.

Regarding to effect of C_{FN} on the device performance, this is discussed in separate places in the manuscript as well shown as data in Table 1. For TFT applications, low-coverage fiber network is optimum since additional packed fibers do not allow efficient switching. For resistor-based devices, middle/high density is preferred since this enhances conductivity and, for sensors, sensitivity due to high surface area. High-coverage fiber networks are optimum for self-standing samples, which can be more easily integrated into monolithic devices. In revision, we have added a schematic in the SI to clearly define the fiber coverage (new Figure S1b), revised the caption of Fig. S1 to emphasize the effects of fiber density, and give the definition of fiber density on in page 3 of the revised text.

2: The authors stated that “Figure 2c shows the $(IDS)^{1/2} \sim VGS$ curves for these IGZO TFTs exposed to different gaseous NO_2 concentrations at $25^\circ C$ ”. However, in Figure 2c, the y-axis indicates IDS (mA). It is recommended to check the Figure 2c.

Response: We thank Reviewer 2 for pointing this out. This typographical error that has been corrected in revision.

3: The authors stated that “The calculated R_{ph} , on/off current ratio (I_{ph}/I_d), D^* at a voltage of 5 V, and response time are shown in Figure 4c”. However, the response time of IGZO photodetector is not described in Figure 4c. It is recommended to check Figure 4c.

Response: We thank Reviewer 2 for pointing out this typographical error. The response time was given in Figure S16 (SI) and, in revision, is corrected on Page 12.

4: The authors claimed that “The typical fabrication consists of dissolving the metal salt(s) (100-200 mg/mL) in ethanol, followed by the addition of an appropriate amount of polyvinyl butyral (PVB; 100 mg/mL) and stirring the solution for 2 h. The use of this particular polymer is found to be critical”. But the reviewer cannot find sufficient information that explains the reason why the use of PVB is critical. The reviewer recommends that the authors to add more explanation on such an argument.

Response: This is an excellent point. There are two reasons why PVB was chosen. 1) It is soluble in low boil-point ethanol and can form fibers by blow-spinning; 2) The addition of metal salts does not affect either the solubility of PVB or fiber formation. Note that several other polymers (like polyvinylpyrrolidone, poly(ethylene oxide)) were not effective. In revision, we now discuss these issues on Pages 4-5.

5: The authors claimed that the conductivities of the IGZO, CuO, ITO, and Cu based devices are remarkably comparable to those of the corresponding bulk materials.

- In Figure S5 figure caption, notations (a)-(d) do not match with corresponding above images.
- Providing detailed conductivities and corresponding references that the authors used to compare proposed devices with the bulk materials would be helpful.
- Some additional statements regarding the comparison with ultrathin single crystal silicon based devices including wearable transistors, circuits, memory devices would be helpful.

Response: These are excellent suggestions. In revision we did the following, 1). Corrected the typographical error in the caption of Figure S5; 2). Added the relevant references (Refs. 6, 31, 32) for the bulk materials on Pages 3 and 5). Added additional statements and references on wearable electronics based on ultrathin single crystal silicon on Page 3.

6: In Figure 2, the authors fabricated TFTs and analyzed their characteristics. However, in case of the integrated wearable device in Figure 4 and Figure 5, the reviewer is curious why the authors used a resistor-type device instead of a TFT-type, in which it can provide better performances in terms of the sensitivity.

Response: Excellent point. We had attempted to fabricate the transistor-based gas sensors in preliminary experiments, however, the TFT performance was very poor when replacing SiO₂/Si with an elastomeric dielectric+ substrate. Since the present resistor work is already very comprehensive, we feel that optimization TFT-type sensors is more appropriate for a future publication.

Reviewer: 3

Comments:

This work proposed the efficient production at scale of several metal oxide and metal nanofibers and fiber networks (FNs) using a simple blow spinning technique employing formulations of metal salt(s) and a polymer, which shows the novelty and inspiration in this field. However, the following questions need to be addressed item by item so as to give further acceptance considerations (Major Revision).

We greatly appreciate the acknowledgement by the Reviewer of the novelty and inspiration of this work.

1. Why did you respectively choose PI and SEBS as the substrate for IGZO FN-based devices?

Response:

PI question. Polyimides are widely used to fabricate flexible electronic devices because of their lightweight, biocompatibility, solution processability (from PI precursors), as well as inherent high bendability and toughness. Polyimides are also some of the most thermally stable and chemically resistant plastic substrates available, and hence are suitable for photolithography and direct physical vapor deposition of metals without inducing significant stresses in the substrate. In addition to the above advantages, PI adheres strongly to the transferred fiber network without additional bonding agent.

SEBS question. Although elastomeric polymers like PDMS and PU have been used for stretchable devices, we used SEBS here because it exhibits superior elongation (>600%) and tenacity.

In revision, we have now clarify both points and cited a relevant reference (no. 45) on Page 9.

(2) Please point out the differences between PEDOT: PSS/[EMIM] [TFSI] and Cr/Au as IGZO FN-based resistor electrodes, do not only mention flexibility.

Response: The main differences between these materials are the conductivity and stretchability. Although the conductivity of PEDOT: PSS/[EMIM] [TFSI] is significantly lower than that of Cr/Au (~ 170 S/cm vs. 4.1×10^5 S/cm), the former films are far more stretchable, as reported in several studies (e.g. *Science Advances* **2017**, 3, e1602076). To demonstrate that the present IGZO FN-based devices are highly stretchable ($\sim 50\%$), the elastic deformation of the measurement electrode must be large so that the electrode conductivity is not compromised during large strains. From the literature and our own experience, Cr/Au electrodes crack during larger deformation. In revision, this point is now discussed on Page 11.

(3) Please explain how IGZO FN-based resistor distinguishes strain, light, temperature, gas and humidity.

Response: For the IGZO FN-based devices, the resistance increases upon exposure to NO_2 gas or when the device is strained, while the resistance decreases upon ambient light exposure, or by increasing the temperature, or by increasing the humidity. With regard to the NO_2 gas and strain stimuli, we plot the gauge factor (GF) and gas sensitivity (S_G) of IGZO FN-based devices under 0%, 10%, and 50% strain below in Figure C1. The GF is far smaller than S_G at 10% strain, indicating the NO_2 gas affect the resistance of devices under small strain. However, when the strain reaches 50%, an opposite trend is observed. Thus we can easily distinguish strain from NO_2 gas. Note, in actual use in wearable devices, the skin strain from the fingers, forehead, etc. is usually less than 10%, and thus the strain within such limit has negligible effect on the NO_2 gas detection.

In terms of light, temperature, and humidity, the sensitivity is $\sim 16 \text{ mA W}^{-1}$, $\sim 2.1\%/^\circ\text{C}$, and $\sim 2\%/RH$, respectively. Thus this device is very sensitive to UV light, thus it can be distinguished easily from other stimuli. However, the differentiation between temperature and humidity requires the integrated e-skin platform (IGZO+ITO+CuO) we described.

In revision, these points are now discussed in greater depth on Page 21 of SI.

Figure C1. Gauge factor and gas sensitivity to NO₂ for IGZO FN-based devices under different strain.

(4) Please explain in detail the detection mechanism of IGZO FN-based TFT and IGZO FN-based resistor for gas.

Response: This is another excellent question which, if we understand correctly, is linked to several of Reviewer 3's additional questions below (#5 and #7). For most effects it is possible to explain where the sensitivity originates since they are similar to those reported for other metal oxide based sensors discussed in the literature. Regarding NO₂ detection, since this gas is an oxidizing agent, it is reasonable to assume that it acts as an acceptor of negative charge carriers (electrons) on the surface of the n-type oxide semiconductor, and the conductance decreases via reduction of the overall charge density. In the revision, we addressed this point on Page 8 and cited relevant literature (new Refs. 35,36).

(5) What is the mechanism of IGZO FN-based resistor to detect temperature and humidity.

Response: The mechanisms of the IGZO FN-based resistor to detect temperature (thermally activated electron de-trapping) and humidity (H₂O absorption donates electrons to the oxide lattice) are likely similar to those of other oxide semiconductors. In revision, we have included two sentences in page 13 and a reference (new Ref. 59).

(6) Please explain why the response of IGZO FN-based resistor to gas increases first and then decreases with the increase of strain.

Response: The decrease in resistance for small strain is reasonable due to the enlarged sensing area (greater NF surface for the gas to adsorb) upon applying a 10% strain. However, when the device is strained further (50%), the resistance of the IGZO FN increases because of lost connectivity between the nanofibers. This point is now discussed in the revised manuscript on Page 13.

(7) What is the reason for IGZO FN-based devices can detect human alcohol consumption.

Response: The mechanism is likely similar to that of water since the similarity in the chemical structure (H-O-H vs. Et-O-H), with a decrease of the resistance upon exposure. The reduction is lower compared to water probably because the acidity of water is greater than that of ethanol and, as shown in Figure 4h, new small peaks appear during the inhalation/exhaling breathing cycle due to the different binding processes of EtOH vs. H₂O. However, these small peaks are important indicators of alcohol consumption since they do not appear when the device is exposed to normal breath gas. In revision, we now emphasize these aspects on Page 14.

(8) In addition to detecting mechanical strain, whether CuO FN-based devices are also affected by light and temperature. How to distinguish them?

Response: The answer is yes, the resistance of CuO FN-based devices can also be affected by light and temperature. Similar to IGZO FN-based devices, we can distinguish them by analysing the resistance's change trends and values. Also, we integrated the e-skin platform with three different devices for further improvement. Furthermore, for practical use, each commercially available sensor has its work conditions (like temperature, humidity, light etc.) as the software helps compensate the different effects by regular calibration. Thus, we should only focus on one stimulus for each CuO device and make the compensation for other possible stimuli with software.

(9) Please explain why the resistance of ITO FN-based device increases with the increase of strain.

Response: Excellent point. In the original manuscript (Pages 8-10 and Figure 3) we discussed in detail the morphology/electrical changes of IGZO FN-based resistors by a combination of experiments and simulation. Since both devices are based on the same metal oxide network structure, ITO FN will behave similarly to IGZO FN in terms of charge transport response to mechanical deformation. In revision, we included a note on Page 15.

(10) As regard to the IGZO FN-Based Thin-Film Transistors and Gas Sensors section, you should supplement the following experiments:

(a) What is the selectivity of the gas sensor toward different reducing gases and different VOCs?

Response: As reported in Figure 4e, we have explored the gas selectivity of the resistor-based IGZO FN and found it is not sensitive to reducing gases (like NH_3 and H_2) in the test range (20-1000 ppm). Since both devices are based on the same material, we expect no response. Regarding testing VOCs, we feel that the data we reported here are more than sufficient to demonstrate the points we want to convey and feel that going into testing VOCs, as well as other important analytes such as explosives, biomarkers, and toxins can be the topic of further publications.

(b) How to exclude the influence of different humidity on the performance of gas sensors?

Response: Thank you. Actually, to answer this question would require years of work to achieve (collective) calibration curves and develop proper compensation circuits. If the Reviewer means specifically to NO_2 , we reported that for a IGZO-FN resistor operated at given voltages, increasing the humidity increases the current, while exposure to NO_2 leads to a decrease in current (Figure S19, Figure 4). Thus all experiments (IGZO TFTs and resistors) related to gas exposure done in this work were set at RH~50% to mimic the actual environmental conditions (please see experimental section of Page 19 for details). In revision, we clarified this issue on page 8.

(c) How to explore the best operating temperature and the best response/recovery time?

Response: For NO_2 , the best response/recovery time (2/20 s) obtained in this work is far better than those (100 s-5 min) reported for thin-film metal oxide gas sensors, regardless of the operating temperature (RT-500 °C) and those (~50 s/>2 min) of commercially available NO_2 gas detectors (operated at >300 °C). Note that ~50 s response time meets the requirement for actual use but our devices have response time of only 2-3 s. Hence, we selected to operate our device at room temperature, which is the best choice to minimize energy consumption. To further explore the improvement in NO_2 gas sensing at room temperature using IGZO FN, future studies can be conducted to investigate material structure-composition-operating conditions-sensitivity correlations. However, that is out of our scope of this paper. For details, please see the references and websites: Scientific Reports, **2019**, 9, 7459. *J. Mater. Chem. C*, **2019**,7, 8616-8625; *Chem. Mater.* **2005**, 17, 15, 3997-4000; *Sensor Actuat B-Chem* **2009**, 141(1): 239-243; <http://www.alphasense.com/index.php/products/nitrogen-dioxide/> <https://teledynegasandflamedetection.com/en/gases/no2>.

In revision, we now add a comment on this subject on Page 8.

(d) What's the reproductivity of the gas sensor from batch to batch, not for different device in same unit?

Response: Excellent point. A great deal of time was spent optimizing the experimental conditions from the preparation of precursors, to blow-spinning, and further in the annealing protocol. At least five devices based on different batches were evaluated for the gas sensors and the values we reported are the average range. The reproducibility will be good if the experimental procedures reported in this paper are followed. In revision, we have now provided the reproducibility data (Figure S7, S8 and S17) on Pages 8-9 and 18 in the SI for IGZO FN-based TFTs and gas sensors and in a footnote under the Table 1 on Page 7.

(11) 10% strain interval is not strict, you should show experimental data of 5% strain interval.

Response: A good point. For revision, another set of experiments was conducted and the resistance change of the IGZO FN devices with a 5% strain is now reported in Figures 3h and 3i on Page 10.

(12) Only 1000 strain cycles are not sufficient.

Response: In revision we now include 5000 stretching cycles tests for the IGZO FN-based devices; see revised Figure 3j on Page 10.

(13) You mention that the device exhibits very short response/recovery times (40/60 ms) when a finger touches/leaves the CuO FN/SEBS device, How do you determine this time and what equipment do you use for testing, you need to reflect more detailed information in Figure 4-l.

Response: Thank you for pointing out this. The equipment employed was a B1500A semiconductor analyzer using the *I-t* setup supplied with the software. The current changes were measured with a sampling interval of 20 ms at given voltages when the index finger touched/left the CuO FN/SEBS device quickly. In revision we have now added this detailed information in the Experimental Section on Pages 20.

(14) Some experimental images need to be re-polished in order to improve the quality of the paper.

Response: Thank you. In revision, we revised all the Figures by rearranging sizes and positions the plots, making them not stick together. Note, these figures were originally supplied with lower resolution to avoid a large file.

We hope this revised version of our work is acceptable and that the paper can be published.

All the best.

REVIEWERS' COMMENTS:

Reviewer #1 (Remarks to the Author):

The paper is properly revised. I recommend the paper is published as it is.

Reviewer #2 (Remarks to the Author):

All comments from the reviewer were well addressed in the revised manuscript.

Reviewer #3 (Remarks to the Author):

The author has given detailed responses according to each review comment of the reviewer, and has now met the publishing requirements.

Reviewer #1 (Remarks to the Author):

The paper is properly revised. I recommend the paper is published as it is.

Response: Thank you for reviewing the paper.

Reviewer #2 (Remarks to the Author):

All comments from the reviewer were well addressed in the revised manuscript.

Response: Thank you for reviewing the paper.

Reviewer #3 (Remarks to the Author):

The author has given detailed responses according to each review comment of the reviewer, and has now met the publishing requirements.

Response: Thank you for reviewing the paper.